# Topological magneto-optical Kerr effect without spin-orbit coupling in spin-compensated antiferromagnet

Camron Farhang[1,4], Weihang Lu[1,4], Kai Du[2], Yunpeng Gao[3], Junjie Yang[3], Sang-Wook Cheong[2] & Jing Xia[1] ✉

The magneto-optical Kerr effect (MOKE), the differential reflection of oppositely circularly polarized light, has traditionally been associated with relativistic spin-orbit coupling (SOC), which links a particle's spin with its orbital motion. In ferromagnets, large MOKE signals arise from the combination of magnetization and SOC, while in certain coplanar antiferromagnets, SOC-induced Berry curvature enables MOKE despite zero net magnetization. Theoretically, large MOKE can also arise in a broader class of magnetic materials with compensated spins, without relying on SOC - for example, in systems exhibiting real-space scalar spin chirality. The experimental verification has remained elusive. Here, we demonstrate such a SOC- and magnetization-free MOKE in the noncoplanar antiferromagnet $Co_{1/3}TaS_2$. Using a Sagnac interferometer microscope, we image domains of scalar spin chirality and their reversal. Our findings establish experimentally a new mechanism for generating large MOKE signals and position chiral spin textures in compensated magnets as a compelling platform for ultrafast, stray-field-immune opto-spintronic applications.

The magneto-optical Faraday[1] and Kerr[2] effects, which reflect the fundamental interactions between circularly polarized light and magnetic materials, have been known for over a century. Among them, the magneto-optical Kerr effect (MOKE) has been extensively utilized for probing magnetic properties[3], visualizing magnetic domains[2], manipulating magnetic order[4], and enabling a variety of magneto-optical technologies[5], even down to the two-dimensional (2D) limit[6,7].

Spin-orbit coupling (SOC)[8,9], a relativistic interaction between a particle's spin and its orbital motion, has been deemed central in generating large MOKE signals. The presence of MOKE in ferromagnets has been attributed to the interplay between SOC and band exchange splitting (BES) in the band structure[10] induced by either the spontaneous magnetization or an external magnetic field. In non-collinear antiferromagnets with negligible net magnetization[11], large MOKE signals up to 300 μrad have recently been predicted[12] and observed[13],

due to a non-zero net Berry curvature in the band structure induced by SOC. This dependence on SOC appears to be a general requirement: although finite MOKE can be observed in systems such as time-reversal symmetry-breaking superconductors[14–16] and orbital Hall materials[17] with negligible SOC, the magnitude of the effect in these cases is typically several orders of magnitude smaller, often in the nanoradian range. With the emergence of novel magnetic materials such as altermagnets[18,19], where SOC and net magnetization are both intrinsically weak or absent, it becomes a question of fundamental significance whether large MOKE signals can be achieved without relying on SOC or net magnetization. This is also of practical importance, as MOKE offers a practical readout for spintronic and opto-spintronic devices based on altermagnets and antiferromagnets of compensated spins, with minimal stray fields, field robustness, and fast switching[20–23].

[1]Department of Physics and Astronomy, University of California, Irvine, Irvine, CA, USA. [2]Keck Center for Quantum Magnetism and Department of Physics and Astronomy, Rutgers University, Piscataway, NJ, USA. [3]Department of Physics, New Jersey Institute of Technology, Newark, NJ, USA. [4]These authors contributed equally: Camron Farhang, Weihang Lu. ✉e-mail: xia.jing@uci.edu

Theoretically, spin-dependent band splitting, and thus MOKE, can also arise from real-space spin textures of compensated spins without referencing relativistic SOC[24–26]. For instance, non-relativistic exchange-driven band splitting occurs in itinerant electron systems, giving rise to phenomena like altermagnetism[18,19]. One of the simplest examples is a noncoplanar antiferromagnet, proposed to exhibit a novel topological light-matter interaction[24], inducing a topological MOKE signal due to finite scalar spin chirality without relying on SOC or BES, i.e., without a net magnetization. In its minimal configuration as illustrated in Fig. 1a, three noncoplanar twisted spins $S_i$, $S_j$, and $S_k$ generate a fictitious U(1) gauge field $b_f \propto t_3 \chi_{ijk} \hat{n}$, where $t_3 = t_{ij} t_{jk} t_{ki}$ is the product of successive electron hopping integrals around the triangular loop $i \to j \to k \to i$, $\chi_{ijk} = S_i \cdot (S_j \times S_k)$ is the scalar spin chirality representing real-space "spin-winding", and $\hat{n}$ is the unit vector normal to the face of the triangle. This fictitious field $b_f$ originates from the orbital motion of electrons: as an electron hops around the triangle, it acquires a geometric Berry phase[27] corresponding to the solid angle subtended by the three spins. This topologically nontrivial Berry phase results in a phase difference $\Delta\varphi$ between left- and right-circularly polarized (LCP and RCP) light upon reflection, manifesting as a Kerr angle $\theta_K = \Delta\varphi/2$. Notably, this coupling between spin and orbital degrees of freedom (spin-dependent band splitting) arises from Kondo exchange coupling between conduction electrons and a noncoplanar arrangement of localized magnetic moments, without requiring relativistic SOC[25].

Despite the promising potential of this SOC-free mechanism for MOKE in compensated magnetic systems, experimental verification has remained elusive. Proposed candidate systems such as $\gamma$-$Fe_xMn_{1-x}$ and $\gamma$-$Na_xCoO_2$[24] have not exhibited the expected signatures. This absence is partly due to the limited exploration of non-coplanar antiferromagnetic orders and partly because the fictitious magnetic field arising from scalar spin chirality often cancels out due to symmetry, typically requiring external strain to stabilize a net effect[24]. Lastly, the choice of the optical wavelength is non-trivial as the theoretically calculated Kerr signal oscillates strongly with the photon energy, crossing zero frequently[24].

In this study, we report the first experimental realization of a large spontaneous MOKE signal of 250 $\mu rad$ at the technologically relevant telecommunication wavelength of 1550 $nm$ in the triangular lattice compound $Co_{1/3}TaS_2$[28–30], despite its negligible net magnetization of 0.01$\mu_B/Co$. This result confirms the viability of the SOC-free mechanism for producing technologically relevant MOKE signals. Using MOKE imaging, we visualize scalar spin chirality domains and their switching behavior under applied magnetic fields.

## Results

$Co_{1/3}TaS_2$ belongs to a broader family of magnetic element-intercalated van der Waals transition metal dichalcogenides, which exhibit a wide range of magnetic states depending on the intercalants[31–36]. Its crystal structure consists of chiral alternating layers

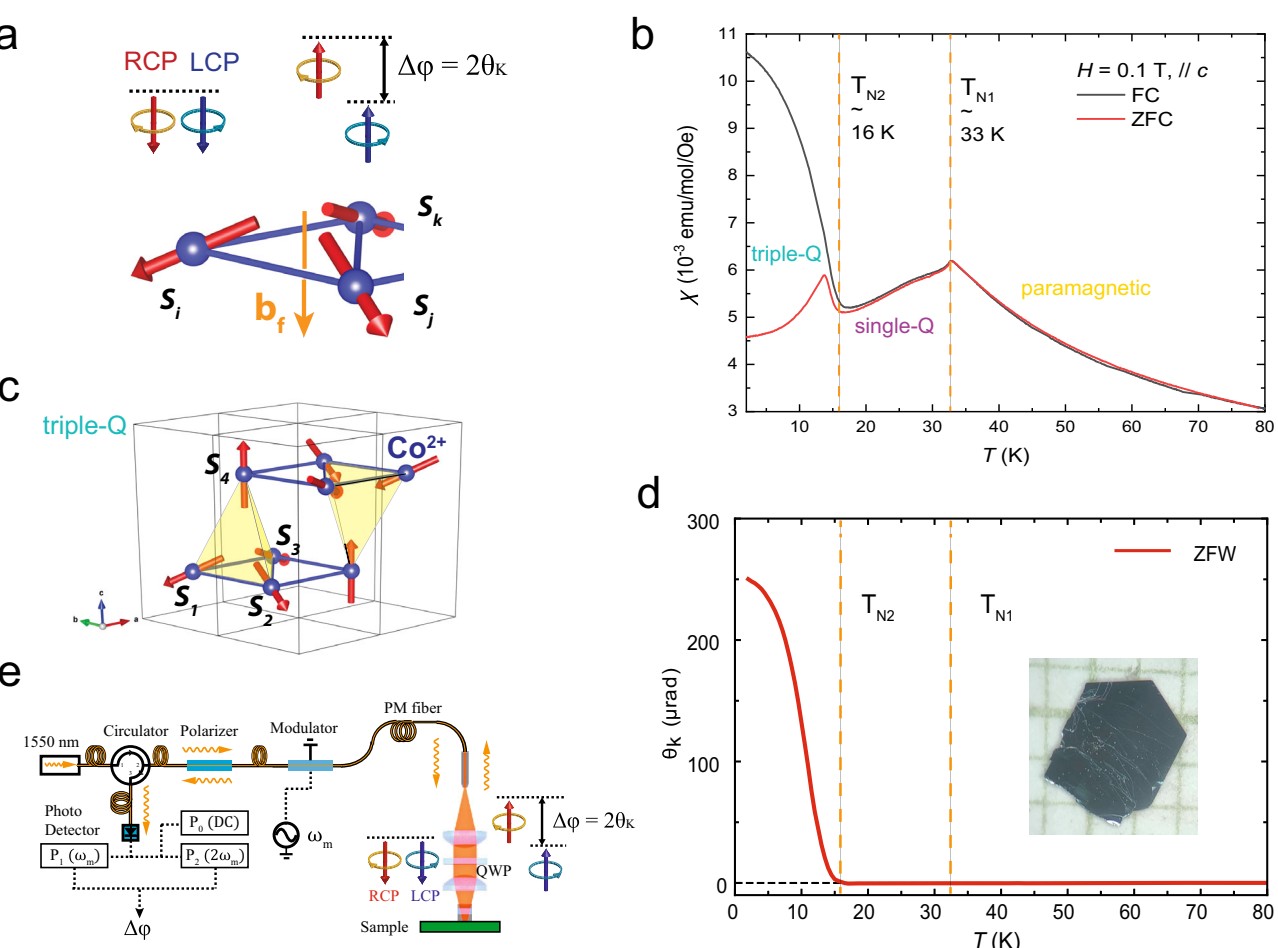

**Fig. 1 | Large MOKE in non-coplanar triple-Q state of $Co_{1/3}Ta_1S_2$. a** MOKE due to a new topological light-matter interaction in a minimal chiral magnet comprising three neighboring noncoplanar spins. $b_f$ is the fictitious magnetic field generated by the scalar spin chirality $\chi_{ijk} = S_i \bullet (S_j \times S_k)$. MOKE $\theta_K$ is a result of the phase difference $\triangle\varphi = 2\theta_K$ between reflected left and right-circularly polarized (LCP and RCP) light. **b** Magnetization $M_z$ measured after field cool (FC) and zero-field cool (ZFC). **c** The tetrahedral triple-Q state of $Co_{1/3}Ta_1S_2$. **d** MOKE $\theta_K$ measured during ZFW after 0.3 $T$ FC. Inset is a sample photo on 1$mm$ grid paper. **e** Schematics of a zero-area-loop Sagnac interferometer microscope for polar MOKE measurements.

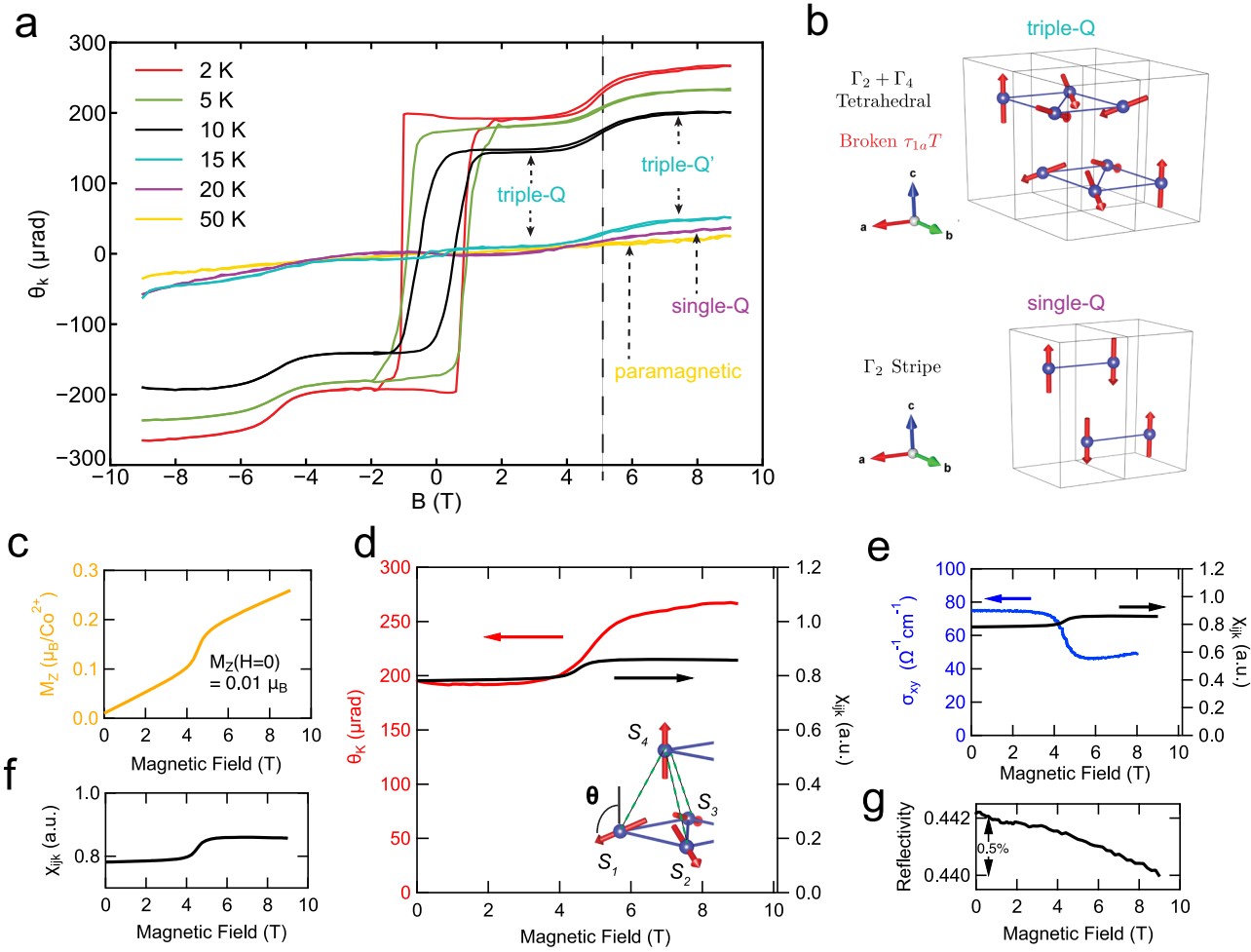

**Fig. 2 | MOKE hysteresis. a** MOKE hysteresis taken at spot 1 at various temperatures. **b** Sketches of triple-Q and single-Q phases. **c** Magnetization $M_z$ at $2K$. **d** MOKE at $2K$ plotted with the estimated spin chirality $\chi_{ijk}$. **e** Hall conductivity $\sigma_{xy}$ at $2K$ plotted with $\chi_{ijk}$. **f** Spin chirality $\chi_{ijk}$ estimated from magnetization. **g** Reflectivity at $2K$.

of cobalt, which carry localized magnetic moments, and metallic tantalum disulfide, which hosts itinerant electrons. While neutron scattering studies in the 1980s reported[33] a magnetic modulation vector of $q_m = (1/3, 1/3, 0)$, leading to interpretations involving a "toroidal" magnetic order[28], recent experiments[29,30] on $Co_{1/3}TaS_2$ and $Co_{1/3}NbS_2$ have instead identified $q_m = (1/2, 0, 0)$. The latter corresponds to a non-coplanar "triple-Q" order[29,30], simultaneously breaking time-reversal and inversion symmetry. Consequently they exhibit anomalous Hall effect[29,30,37] and Nernst effect[38] arising from an uncompensated fictitious magnetic field, even though their net magnetic moments are vanishingly small. A clear correlation between the degree of spin non-coplanarity and the amplitude of spontaneous Hall signal was confirmed by temperature dependence of polarized neutron scattering[30]. As illustrated in Fig. 1c, the magnetic unit cell in this triple-Q state includes eight triangular plaquettes that generate a strong fictitious field oriented along the crystallographic c-axis. Importantly, this magnetic order in an ideal triple-Q state[29] does not involve any net spin magnetization; rather, the fictitious field couples to the orbital motion of itinerant electrons, producing a small orbital magnetic moment of approximately $0.01\mu_B$ per $Co^{2+}$ ion[29,30]. Note that the real material $Co_{1/3}TaS_2$ may deviate from this ideal triple-Q state, and the observed zero-field magnetization could include some spin contribution.

This spin configuration has non-zero scalar spin chirality and is not associated with any spin rotation symmetry, so non-relativistic spin splitting can occur along any direction. In addition, this spin

configuration can have a net magnetization along the c-axis when SOC is included, so it may be considered as an "M-type" altermagnet with a magnetic point group of 32′[39].

Upon warming, this triple-Q order transitions at the Néel temperature $T_{N2}$ into a single-Q stripe order as illustrated in Fig. 2b with zero scalar spin chirality, and subsequently into a paramagnetic state above $T_{N1}$[29]. The $Co_{1/3}TaS_2$ crystals used in our experiments were synthesized via chemical vapor transport (Methods), as shown in the inset of Fig. 1d. Magnetic susceptibility measurements under field-cooled (FC) and zero-field-cooled (ZFC) conditions (Fig. 1b) indicate transition temperatures of $T_{N1}$~33 $K$ and $T_{N2}$~16 $K$.

MOKE detection and imaging are performed using a zero-loop Sagnac interferometer microscope[6,14,40,41] in the polar geometry (Fig. 1e) operating at the most popular telecommunication wavelength of 1550$nm$ (0.80$eV$ photon energy). This technique offers exceptional sensitivity, routinely achieving 0.01$\mu rad$ resolution. This is made possible by its exclusive detection of microscopic time-reversal symmetry breaking (TRSB) while strongly rejecting non-TRSB effects, such as optical birefringence, with a suppression level of $10^{-6}$ (Supplementary Information). This distinction is particularly important in $Co_{1/3}TaS_2$, where some of the authors have recently observed a birefringent polarization rotation of 600$\mu rad$ sin(2$\alpha$) due to a nonvolatile nematic order, with $\alpha$ being the incident polarization angle[42]. The selective sensitivity to TRSB is achieved by using a single-mode optical fiber as both the source and detector for counter-propagating, time-reversed light beams[43]. According to Onsager's

relations[44], this configuration guarantees zero signal in the absence of TRSB.

We now present the results of our MOKE measurements. Figure 1d shows the spontaneous MOKE signal $\theta_K$ obtained during zero-field warming (ZFW) after $B = 0.3T$ field cooling (FC), measured at a single point with $2\,\mu m$ optical beam size. Remarkably, we observe a giant $\theta_K$ of $250\mu rad$, comparable to the $300\mu rad$ signal reported in the coplanar non-collinear antiferromagnet $Mn_3Sn$[13], where the large MOKE arises from relativistic SOC-induced Berry curvature[12]. It is important to note that both $Co_{1/3}TaS_2$ and $Mn_3Sn$ exhibit vanishingly small net magnetic moments, with values of $\sim 0.01\mu_B/Co$[29,30] and $\sim 0.005\mu_B/Mn$[45], respectively. This experimental result thus firmly establishes that the SOC-free mechanism driven by the scalar spin chirality can produce MOKE signals on par with those generated by SOC-induced Berry curvature. The pronounced temperature dependence of MOKE in $Co_{1/3}TaS_2$ stands in sharp contrast to the temperature-invariant behavior we observed in the coplanar non-collinear antiferromagnet $Mn_3NiN$[46], underscoring their distinct origins. When compared to theoretical predictions, the calculated $\theta_K$ in the proposed triple-Q state of $\gamma$-$Fe_xMn_{1-x}$ is even larger, reaching $\pm 10 mrad$ depending on the photon wavelength[24]. These observations suggest that the SOC-free mechanism holds substantial promise for approaching the largest reported MOKE of ~10 $mrad$ in ferromagnet $CoPt$[47].

In contrast to the large $\theta_K$ in the triple-Q state, no measurable MOKE signal was detected during ZFW through $T_{N2}$ into the single-Q phase, or upon further warming above $T_{N1}$ into the paramagnetic phase (Fig. 1d). This is consistent with theoretical predictions: both the single-Q and paramagnetic phases lack scalar spin chirality and thus are not expected to support a finite MOKE[24].

We observed negligible spontaneous $\theta_K$ after zero-field cool (ZFC) (Supplementary Fig. 7b), indicating the formation of oppositely polarized domains smaller than the optical beam size in the absence of a training field. This also confirms that the zero-field MOKE signal does not originate from uncompensated spins at AFM domain walls. If that were the case, ZFC, which produces a higher density of domain walls, would yield a larger MOKE signal than field cooling (FC), contrary to our observations. The negligible magnetic contribution from AFM domain walls is also supported by MFM measurements after ZFC (Supplementary Fig. 8), which show no detectable fringing fields from domain walls.

Figure 2a summarizes the MOKE hysteresis loops across different magnetic phases of $Co_{1/3}TaS_2$. At $50K$, in the paramagnetic phase, the Kerr signal $\theta_K$ exhibits a small linear response to the applied magnetic field $B$, consistent with partial alignment of magnetic moments by the external field. At $20K$, in the single-Q phase as illustrated in Fig. 2b, the MOKE signal develops a shallow S-shaped curve but remains zero at zero field, consistent with the absence of scalar spin chirality $\chi_{ijk}$. Upon cooling to $15K$, as the system enters the non-coplanar triple-Q phase, a small hysteresis loop emerges between $\pm 0.4T$ (cyan curve in Fig. 2a), signaling the onset of a finite $\chi_{ijk}$ and the associated fictitious field $b_f$. Notably, across $\sim 5T$ magnetic field, there is a smooth change in the Kerr signal, corresponding to a metamagnetic transition into a different magnetic configuration[29,30], referred to here as the triple-Q' state. Further cooling to $10K$, $5K$, and $2K$ leads to increasingly wide and pronounced hysteresis loops, indicating higher coercive fields required to reverse the chirality domains and stronger fictitious fields at zero applied field. The jump in MOKE signal across $5T$ field also becomes more abrupt at lower temperatures, signifying a sharper metamagnetic transition between the triple-Q and triple-Q' phases.

The most striking feature of the Kerr hysteresis in the triple-Q phase (Fig. 2a) is its field independence: once the sign of $\chi_{ijk}$ is established by the external field, the Kerr signal remains nearly constant between $0T$ and $4T$ fields. A similar plateau is observed in the triple-Q' phase above $6T$. For clarity, the $2K$ MOKE hysteresis is replotted in Fig. 2d, revealing a stark contrast to the corresponding $2K$

magnetization $M_z$ (Fig. 2c). $M_z$ begins at $\sim 0.01\mu_B$ per Co at zero field and increases linearly at a rate of $dM_z/dB \sim 0.02\mu_B/T$, except for a discrete jump of $\sim 0.07\mu_B$ at $5T$ due to the metamagnetic transition. We estimate that the magnetization contribution to the MOKE signal is negligible, less than 1% (see Supplementary Information). This contrasting field dependence between MOKE and $M_z$ confirms their decoupling: while MOKE arises from the fictitious field $b_f$ generated by scalar spin chirality, the magnetization includes both spin and orbital contributions. Specifically, the zero-field $M_z$ and its linear slope $dM_z/dB$ are due to the orbital moment of itinerant electrons, induced respectively by the fictitious field $b_f$ and the external magnetic field $B$. These orbital contributions are not directly coupled to the Kerr signal. In contrast, the metamagnetic transition, which involves a spin reconfiguration, produces simultaneous step-like changes in both $M_z$ and $\theta_K$. Importantly the optical reflectivity remained constant to within 0.5% during the entire measurement (Fig. 2g), confirming that the metamagnetic transition represents a change of the spin configuration and has minimum impacts on the electronic structure.

Since the nature of the metamagnetic transition has not been resolved by neutron scattering experiments yet[29,30], we adopt a simplified working assumption: the transition reflects a sudden change in the angle $\theta$ between spins $S_1$ and $S_4$, as illustrated in the inset of Fig. 2d. Under this assumption, the net spin moment is given by $M_{spin} = \frac{1}{2}(-1 - 3\cos(\theta))\mu_B$, assuming a g-factor of -2. And it can be estimated experimentally from magnetization as $M_{spin} \sim M_z - 0.02\mu_B \cdot B$, accounting for the orbital contribution. The scalar spin chirality $\chi_{ijk}$ is assumed to scale with the bottom triangular plaquette $\chi_{ijk} \propto \chi_{123} \propto -\frac{3}{2}\sqrt{3}\cos(\theta)\sin(\theta)^2$. Within this simplistic model, we estimate $\chi_{ijk} \sim -\frac{2(1 + 2M_{spin})(-2 + M_{spin} + M_{spin}^2)}{3\sqrt{3}}$, as plotted in Fig. 2f. Comparing this estimated $\chi_{ijk}$ and the measured $\theta_K$ (Fig. 2d), the model qualitatively captures the key features of the magnetic-field dependence of MOKE. The quantitative discrepancy in the transition across $5T$ suggests that a more accurate model is required for this metamagnetic transition. This will depend on resolving the spin structure of the triple-Q' phase in future neutron scattering experiments, beyond the scope of the present work. Lastly, we note the DC Hall effect $\sigma_{xy}$ (Fig. 2e) shows an opposite change across the metamagnetic transition, suggesting a rather complex frequency dependence related to the corresponding change of spin texture across the metamagnetic transition.

Spin chirality can be imaged by MOKE microscopy that provides a powerful, non-contact method for mapping spin chirality domains and their associated dynamics. One immediate application is the assessment of chemical inhomogeneity. Recent neutron scattering and transport studies[48] have reported that even slight variations in cobalt composition can lead to significant changes in the physical properties of $Co_xTaS_2$. It is therefore critical to determine the extent of chemical inhomogeneity, and its impact on the MOKE signal within a single crystal. We found that higher Co content leads to lower red-light reflectivity, an effect which we expect to persist at 1550 $nm$ in the Sagnac measurements. (Supplementary Fig. 9 c, d).

In Fig. 3, we present a combined study of optical reflectivity imaging, MOKE imaging, and single location MOKE hysteresis, all performed at $2K$. The optical reflectivity map (Fig. 3b) is largely uniform with $0.42 < R < 0.44$, which we use as a convenient proxy for cobalt composition as it is difficult to perform energy dispersive x-ray spectroscopy (EDX) chemical mapping in the same region. MOKE hysteresis loops measured at a few representative locations with varying reflectivity values are shown in Fig. 3a. All locations exhibit nearly identical coercive fields of $\sim 0.8T$, with the primary variation being in the zero-field Kerr signal (200, 202, 160, and $80\mu rad$, respectively). To further examine the magnetic inhomogeneity, we performed MOKE imaging at three key points along the hysteresis loop: at zero field (Fig. 3d), and near the switching fields at $-1T$ (Fig. 3c)

and +0.8 T (Fig. 3e). Near the switching fields, large variations in both positive and negative Kerr signals are observed (Fig. 3c, e), consistent with the coexistence and reversal of spin chirality domains across the sample. Importantly, the spatial patterns seen in Fig. 3c, e do not resemble those in the reflectivity map (Fig. 3b), suggesting that the coercive field and domain switching behavior are not strongly correlated with cobalt composition. Instead, they are likely influenced by local strain, defects, or other extrinsic structural factors.

In stark contrast, a clear correlation emerges between the optical reflectivity (Fig. 3b) and the zero-field MOKE image (Fig. 3d), where the chirality has been uniformly trained to a negative value. This observation implies that cobalt composition primarily affects the magnitude

of the MOKE signal in the zero-field state, rather than the switching dynamics. To further test this hypothesis, we investigated a rare inhomogeneous region (see Supplementary Fig. 1) where reflectivity varies significantly from 0.40 to 0.45. In this region, the spontaneous MOKE signal ranges from $-40\,\mu rad$ to $-180\,\mu rad$. A plot of MOKE vs. reflectivity in Supplementary Fig. 1 reveals an empirical linear relationship within this limited reflectivity range, supporting the idea that local cobalt concentration modulates the amplitude of the Kerr signal.

Having established that most of the sample is magnetically uniform, we now demonstrate the utility of MOKE microscopy in visualizing scalar spin chirality domain reversal under an external magnetic field. These measurements were conducted in magnetically

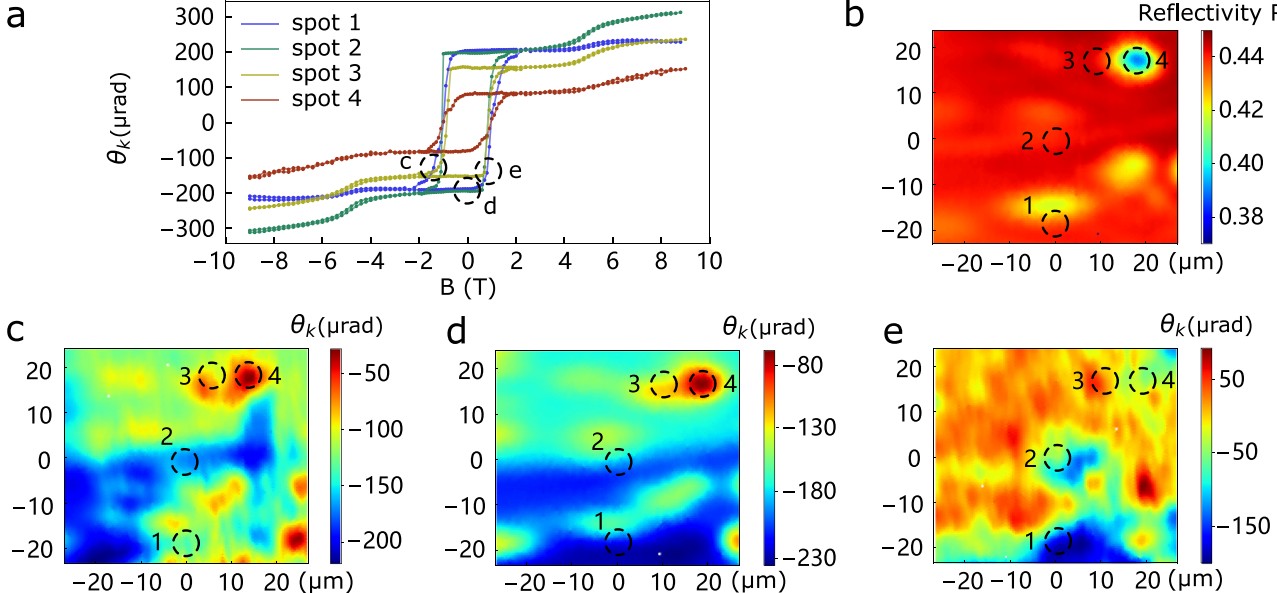

**Fig. 3 | Scalar spin chirality domains imaged at 2K. a** MOKE hysteresis measured at spots 1-4 at $T = 2K$. **b** Optical reflectivity image at $T = 2K$. **c–e** MOKE $\theta_K$ images taken at B = −1 T, 0 T, and 0.8 T respectively during the hysteresis loop at $T = 2K$, showing the formation of scalar spin chirality $\chi_{ijk}$ domains.

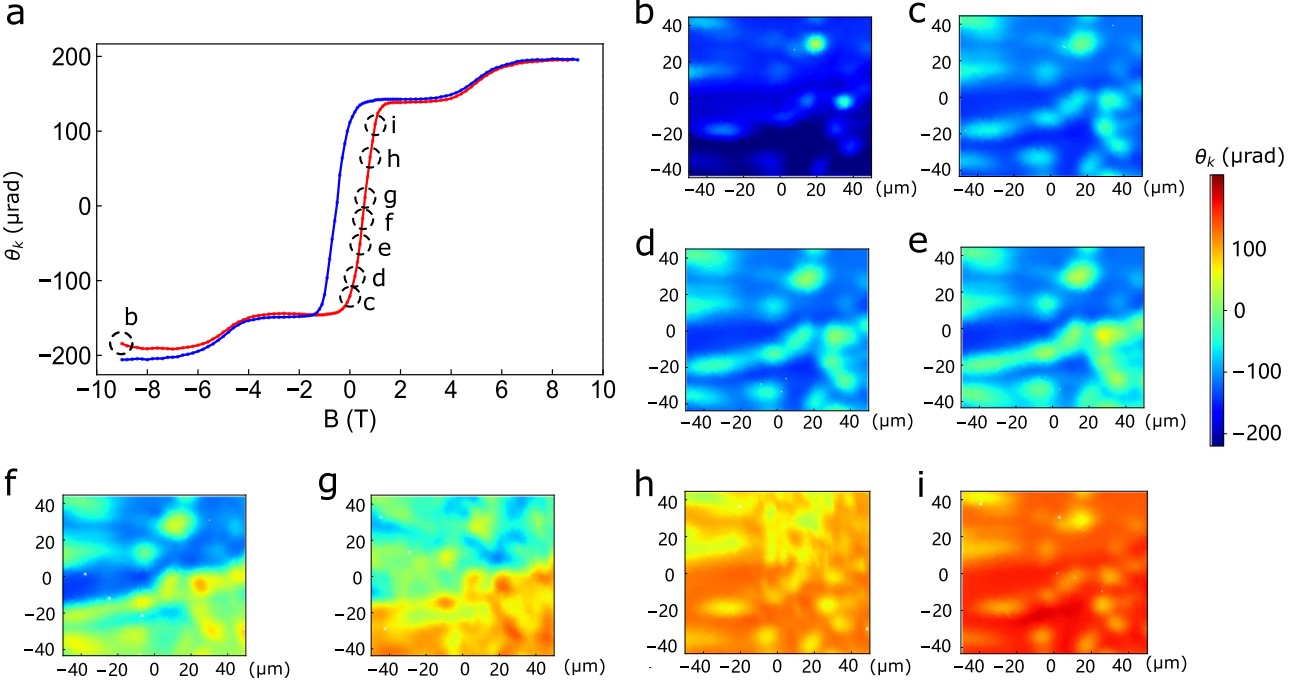

**Fig. 4 | Chirality domain reversal imaged at 10 K. a** MOKE hysteresis measured at a single spot of the imaged region at $T = 2K$. **b–i** MOKE images of chirality domain reversal via domain wall motion at $T = 10K$.

homogeneous regions at an elevated temperature of $10K$, where domain reversal occurs over a broader field range. The results are shown in Fig. 4 and Supplementary Fig. 2.

A sequence of MOKE images from a uniform region is presented in Fig. 4b–i, with the corresponding local hysteresis loop taken at a single location shown in Fig. 4a. At $B = -9T$ (Fig. 4b), the MOKE signal is uniformly negative ($\theta_K \sim -200\mu rad$, shown in blue), except for a few isolated spots that are likely regions of differing Co composition with near-zero Kerr signals (green). At $B = 0$ (Fig. 4c), patches with $\theta_K \sim 0$ begin to emerge. These are interpreted as sub-wavelength domains with mixed positive and negative chiralities, whose contributions average to near zero within the $\sim 2\mu m$ optical spot size. As the magnetic field increases (Fig. 4d–f), these green regions grow and are joined by new areas exhibiting positive chirality (orange to red). The positive domains progressively expand through domain wall motion, eventually dominating the lower half (Fig. 4g) and finally the entire imaged region (Fig. 4h, i), culminating in a uniform positive Kerr signal of $\theta_K \sim 200\mu rad$.

Notably, even at a low field of $0.1T$ (Fig. 4d), isolated positive domains are already present. This indicates that the coercive field for domain reversal is not fundamentally dictated by the intrinsic noncoplanar magnetic phase but is instead governed by extrinsic factors such as local strain, defects, or pinning centers. This insight suggests that by engineering these extrinsic factors, it may be possible to tailor coercive fields in $Co_{1/3}TaS_2$ and related materials, either minimizing them for low-power logic switching applications or maximizing them for robust spin memory devices.

Further evidence of this chirality domain reversal dynamics is provided in Supplementary Fig. 2, which presents MOKE images of another region taken at $1T$ intervals across the full field sweep. As in the main data, domain reversal near $\pm 1T$ occurs via domain wall motion, with positive and negative chirality domains coexisting during the transition.

Interestingly, Supplementary Fig. 2 also captures the distinct dynamics of the metamagnetic transition, which occurs gradually between $B = 4T$ and $6T$. Across the $3T \rightarrow 9T \rightarrow 3T$ cycle, the MOKE signal changes smoothly and uniformly across the field of view, with no evidence of domain wall motion. This behavior resembles a coherent rotation process in ferromagnetic hysteresis, and implies that the critical field for the metamagnetic transition is determined by the intrinsic spin structure transformation. This conclusion is supported by the consistent metamagnetic transition field observed across different samples and temperatures[29,30]. Finally, Supplementary Fig. 3 shows a zero-field MOKE image taken at $20K$ in the single-Q phase. The MOKE signal is uniformly zero across the imaged region, consistent with the expectation that scalar spin chirality, hence MOKE, is absent in the single-Q phase.

## Discussion
The observed topological MOKE in $Co_{1/3}TaS_2$ represents a fundamentally new form of light-matter interaction arising from the chirality of real-space spin textures composed of compensated spin moments, an effect predicted by theory[24] but not previously observed. Unlike conventional MOKE mechanisms that depend on relativistic spin-orbit coupling of individual spins or a net spin moment[10,12], this SOC-free mechanism is rooted in the collective topology of the real-space spin texture with compensated spins. It is therefore expected to apply to a much broader class of magnetic materials with "winding" real-space spin configurations. It is noted that although SOC is not required for the topological MOKE discussed here, it can slightly modify its strength, as shown in theoretical studies of topological Hall conductivity in the DC limit[49].

Supplementary Fig. 4 compares the topological MOKE-over-magnetization ratio ($\theta_K/M$) in $Co_{1/3}TaS_2$ with other magnetic systems. In ferromagnets ($Ni^{50}$, $Fe^{50}$, and $CoPt^{47}$), spontaneous $\theta_K$ scales with remanent magnetization due to SOC, following a $\sim 0.03 rad/\mu_B$ ratio. In contrast, Noncoplanar antiferromagnet $Co_{1/3}TaS_2$ and coplanar antiferromagnet $Mn_3Sn^{13}$ exhibit much larger $\theta_K/M$ ratios due to negligible net magnetization ($0.005 - 0.01\mu_B$). It is important to note that while MOKE in $Mn_3Sn$ is driven by SOC-induced Berry curvature, MOKE in $Co_{1/3}TaS_2$ arises from real-space scalar spin chirality. The skyrmion lattice phase in $Gd_2PdSi_2^{51}$ has a topological contribution to MOKE from swirling spins, though limited to a narrow field range and a large net moment of $4\mu_B$, resulting in a small ratio $\sim 7 \times 10^{-5} rad/\mu_B$. Similarly, small ratios are found in skyrmion lattices in $CrVI_6^{52}$ ($\theta_K \sim 2 \times 10^{-4} rad, M \sim 3\mu_B/Cr$) and graphene/$Fe_3GeTe_2$/graphene heterostructure[53] ($RMCD \sim 1 \times 10^{-3} rad, M \sim 2.5\mu_B/Fe$) primarily due to their large ferromagnetic moments.

Using the noncoplanar spin system in $Co_{1/3}TaS_2^{28–30}$ as a model system, we demonstrated this mechanism's efficacy and employed MOKE microscopy to image both chiral domain reversal and the metamagnetic transition. The observed giant spontaneous MOKE at telecommunication wavelengths is readily detectable using standard experimental setups. Moreover, theoretical work predicts that at terahertz frequencies, this system can host a quantum topological Kerr effect with a quantized Kerr rotation of $\pi/2^{24}$, surpassing all known magneto-optical materials in magnitude.

These findings have direct implications for advancing antiferromagnet and antiferromagnet-based spintronics and opto-spintronics[20–23]. Devices based on this SOC-free mechanism would inherently be immune to stray magnetic fields and capable of ultrafast switching, overcoming key limitations of traditional ferromagnetic technologies. MOKE is well suited for ultrafast, local detection of spin-chirality domain switching driven by chiral spin-orbit torques from electric current or circularly polarized light. More broadly, this mechanism relaxes the design constraints imposed by SOC-based magneto-optical materials, offering greater flexibility in engineering materials with tailored magneto-optical responses. We envision that SOC-free magneto-optical materials, especially the van der Waals layers, can be realized through diverse fabrication techniques, including nanofabrication, molecular beam epitaxy, layer stacking, and self-assembly. This opens a wide and largely unexplored landscape for next-generation magneto-optical materials and devices.

## Methods
### Crystal Growth
$Co_{1/3}TaS_2$ single crystals were grown by chemical vapor transport method. The powder samples of $Co_{1/3}TaS_2$ were synthesized first using the solid-state reaction method. High-purity powders of Cr (99.97%), Ta (99.9%), and S (99.999%) were mixed in stoichiometric ratios. The mixed powders were ground, pelletized, and sealed in quartz tubes. The pellets of powder were then sintered at 750 °C for 48 hours at a heating rate of -15 °C/h, with intermediate grinding. For the single-crystal growth of $Co_{1/3}TaS_2$, the resulting black powder samples were sealed in evacuated quartz tubes together with iodine (I2) as a transport agent. The tubes were placed in a two-zone furnace for 10 days, where the hot and cold ends were maintained at 1000 °C and 900 °C, respectively. The resulting crystals are hexagonal plate-like.

### Magnetization measurements
The magnetic susceptibility versus temperature (χ-T) curves were measured using a Cryogenic-Limited Cryogen Free Measurement System (CFMS) with a vibrating sample magnetometer (VSM) option. A magnetic field of 0.1 T was applied along the c-axis of the $Co_{1/3}TaS_2$ single crystal during the measurements of χ-T curves. The magnetic moment versus magnetic field (M-H) curves were also measured using the VSM option of our CFMS. The crystal was first cooled to base temperature in a zero magnetic field, then the M-H curve was measured by ramping up the magnetic field.

## Hall measurements

The Hall effects of the sample were characterized by a Keithley 2182 Nanovoltmeter and Keithley 6221 Current Source, and the sample temperature and magnetic field were controlled by the CFMS system. During the measurement, an electric current of 1 mA was applied along the ab plane of the crystal, and the magnetic field was applied to the c-axis of the crystal.

## Sagnac MOKE measurements

The MOKE measurements are performed using a zero-loop fiber-optic Sagnac interferometer[43] operating at $1550\,nm$ wavelength. For MOKE imaging we utilize a scanning Sagnac microscope with $2\,\mu m$ lateral spatial resolution installed inside a cryostat with $1.8\,K$ base temperature and $9\,T$ magnetic field capability. The operation of the interferometer is described in the Supplementary Information.

## Low-temperature magnetic force microscopy (MFM)

After zero-field cooling from room temperature, a freshly-cleaved $Co_{1/3}TaS_2$ crystal was scanned at $5\,K$ using a temperature-variable AFM system (Attocube) in a dual pass mode (lift height ~ $35\,nm$) with commercial Co/Cr-coated magnetic tips.

## Data availability

Source data are provided with this paper. They have been deposited in a figshare repository with https://doi.org/10.6084/m9.figshare.29583293.

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

## Acknowledgements

We thank J.G. Zheng for Energy-dispersive X-ray spectroscopy (EDX) at UC Irvine. This project was supported by NSF award DMR-2419425 and the Gordon and Betty Moore Foundation EPiQS Initiative, Grant # GBMF10276 awarded to J.X.; The work at Rutgers University was supported by the DOE under Grant No. DOE: DE-FG02-07ER46382 awarded to S.W.C.; J.Y. acknowledges support by DOE under Grant No. DOE: DE-SC0021188 awarded to J.Y.; The authors acknowledge the use of facilities and instrumentation at the UC Irvine Materials Research Institute (IMRI), which is supported in part by the National Science Foundation through the UC Irvine Materials Research Science and Engineering Center (DMR-2011967).

## Author contributions

J.X. conceived and supervised the project. C.F., W.L., and J.X. carried out the optical measurements. K.D., Y.G., J.Y., and S.W.C. grew the crystals and carried out transport, magnetization, and magnetic force microscopy measurements. J.X. drafted the paper with the input from all authors. All authors contributed to the discussion of the manuscript.

## Competing interests

The authors declare no competing interests.
