## [Peer Review File · Nature Communications]

Topological Magneto-optical Kerr Effect without Spin-orbit Coupling in Spin-compensated Antiferromagnet

Corresponding Author: Professor Jing Xia

Version 0:

Reviewer comments:

Reviewer #1

(Remarks to the Author)

Farhang et al. report the observation of the so-called topological magneto-optical Kerr effect (MOKE) in a compensated antiferromagnet. Using a Sagnac interferometer microscope, they discuss a magneto-optical response arising from spin topology and further argue a correlation between Co composition and MOKE. The experiments appear carefully conducted, and the discussion on composition is intriguing. Although the topological MOKE must be the important topic, I have several concerns regarding the interpretation of MOKE and its novelty, which should be addressed before the manuscript can be reconsidered:

(i) The authors ascribe the zero-field magnetization to an orbital moment induced by a fictitious field in Line 93. Could you clarify why this assignment is justified?

(ii) The authors emphasize the SOC-free mechanism for the MOKE, but I am not convinced their data provide evidence. They show that a simple spin model can explain the anomaly in the magnetic-field dependence of MOKE, but similar anomalies could also arise from SOC-induced MOKE. Whenever the antiferromagnetic structure is modified by an external field, one would expect anomalies in MOKE. Moreover, the field dependence of MOKE differs from that of the topological Hall effect in their data. If both phenomena share the same microscopic origin, their field dependence should be consistent. This discrepancy is critical, and the authors must clarify why the two responses behave differently.

(iii) Related to this point, I strongly recommend the authors measure the wavelength dependence. There should be the particular wavelength where the MOKE shows the same field dependence as the Hall conductivity, which is the genuine topological MOKE.

(iv) The contrast in the optical reflectivity image is attributed to variations in Co composition. Is there direct evidence supporting this assignment? The authors should take an EDX image in the corresponding region.

(v) The authors argue a clear correlation between Co composition and the magnitude of MOKE. The 5 % change in reflectivity results in dramatic change of MOKE signal as shown in Fig. S1. What could be the underlying mechanism? Does the magnetic structure or spin-orbit coupling strength change with Co substitution?

(vi) Topological MOKE has already been reported in several other systems. If the novelty of the present work lies in demonstrating the SOC-free mechanism, its significance is limited. In addition, the authors should provide a balanced discussion of prior studies especially in the introduction and clearly highlight what is genuinely new in their results.

(vii) Even if the MOKE is primarily driven by spin topology, the spin-orbit coupling inevitably plays some role in real materials. It is therefore misleading to stress a purely SOC-free mechanism.

(viii) Figure S4 shows that Mn_3Sn exhibits the largest ratio of MOKE to magnetization. Then, what is the specific advantage of the topological MOKE inferred from the present data?

Reviewer #2

(Remarks to the Author)

In this study, the authors investigate the magneto-optical Kerr effect (MOKE) in the non-coplanar antiferromagnet $Co_{1/3}TaS_2$. They propose that the observed MOKE signal is topological in nature and arises from a mechanism independent of spin-orbit coupling (SOC) and net magnetization. By employing Sagnac interferometry-based MOKE microscopy, the authors simultaneously resolve chiral domain structures and superparamagnetic transition behavior. The work compellingly argues that, in contrast to the conventional SOC-dependent MOKE mechanisms that require either relativistic spin-orbit coupling or a finite magnetization, the signal here originates from collective topological properties of

compensated spins within a real-space spin texture. The findings are intriguing and the data are in high quality. Although the experimental observations are of significant interest, the authors' conclusions require further substantiation. Several concerns regarding the interpretation and experimental design should be addressed prior to publication.

1. The authors attribute the MOKE signal in Co_{1/3}TaS₂ to a mechanism that does not involve SOC or net magnetization. However, certain experimental details appear to indicate a correlation between the MOKE signals and magnetic field-induced changes in magnetization. For example, a comparison between Figures 1b and 1d reveals an increase in the MOKE signal concomitant with a rise in the magnetization M_z . Moreover, an upturn in M_z is observed as the magnetic field is swept from 4 T to 6 T (Figure 2c) due to a transition from a triple-Q to a single-Q state, which is mirrored by a corresponding increase in the MOKE signal (Figure 2d). This suggests that at least part of the MOKE signal may stem from field-induced magnetization. In addition to the data collected during ZFW after 0.3 T FC, could the authors provide MOKE measurements after ZFC?

2. Unlike SQUID, which probes bulk magnetization, Sagnac MOKE is a local measurement technique. Thus, the detected signal may originate not only from the intrinsic crystal properties but also from magnetization at domain walls. Owing to the high sensitivity of Sagnac interferometry microscopy, the authors clearly visualize localized structures in the antiferromagnetic material, as shown in Figures 3 and 4. These images reveal the presence of pronounced magnetic domains within Co_{1/3}TaS₂. At domain boundaries, even in the absence of an external magnetic field ($B=0$), uncompensated magnetization can arise due to spontaneous moments, which could contribute to a MOKE signal at $B = 0$. Therefore, could part of the zero-field MOKE signal be attributable to incomplete spin compensation at domain walls, while the field-dependent component arises from field-induced magnetization (M_z)? To rule out contributions from both uncompensated moments and field-induced magnetization, it is recommended that the authors perform additional local magnetic measurements, such as magnetic force microscopy (MFM), to corroborate their interpretations.

Reviewer #3

(Remarks to the Author)

This work focuses on realizing a magneto-optical Kerr effect (MOKE) in the spin-compensated noncoplanar antiferromagnet Co_{1/3}TaS₂ without relying on spin-orbit coupling, thereby addressing a central open question in the fields of magnetism and light–matter interactions. By employing a highly sensitive zero-loop Sagnac interferometer microscope, the authors achieved direct imaging of scalar spin chirality domains and their reversal driven by noncoplanar spin configurations. They report a giant spontaneous Kerr signal of up to 250 μrad under an applied magnetic field, at a technologically relevant telecommunication wavelength, which is comparable in magnitude to SOC-driven MOKE in noncollinear antiferromagnets (e.g., Mn₃Sn). Importantly, the study also captures the phase transition between 1Q and 3Q magnetic states, revealing that the Kerr signal vanishes in the 1Q phase and emerges robustly in the 3Q phase. Furthermore, the authors provide compelling evidence of chirality domain dynamics, disentangle extrinsic influences such as local strain and defects, and carefully correlate the Kerr response with chemical inhomogeneity. This clear correspondence between spin texture and magneto-optical response underscores the reliability of the experimental observations. The experimental design is meticulous, the methodology is state-of-the-art, and the results constitute the first robust experimental demonstration of the theoretically predicted topological magneto-optical effects.

The significance of this work lies in its demonstration that large, macroscopically measurable "topological" MOKE signals can arise without relying on either spin–orbit coupling or band exchange splitting, thereby fundamentally broadening the current understanding of magneto-optical phenomena. By establishing a direct connection between scalar spin chirality and the optical response—most notably the disappearance of the Kerr signal in the 1Q and paramagnetic phases and its robust reemergence in the 3Q phase—the study provides a well-defined benchmark for identifying and manipulating topological magneto-optical effects. These findings mark a milestone in exploring topological light–matter interactions and open promising avenues for spintronics and opto-spintronics based on chiral magnets, with potential implications for ultrafast, energy-efficient, and stray-field-immune device technologies.

Before publication, I would like to raise two minor comments for the authors' consideration:

1) In the estimation of the scalar spin chirality ($\chi_{\{ijk\}}$), only the bottom triangular plaquette is considered. However, from the crystal structure, there appear to be two layers of spin textures. How does the top triangular plaquette contribute to the scalar spin chirality, and can it qualitatively change the relationship between magnetization and the Kerr signals?

2) In the Discussion section, line 254, there seems to be a typo: Extended Data Fig. 2 should be corrected to Extended Data Fig. 4. In Extended Data Fig. 4, the work on the skyrmion lattice Gd₂PdSi₂ (Ref. 50) is cited and compared with the present results. However, for topological MOKE emerging in skyrmion lattices, two more recent studies should also be cited and discussed: Nat. Phys. 20, 1145–1151 (2024) and ACS Nano 18, 20055–20064 (2024). As this is a rapidly developing and competitive field, including these relevant references and, if possible, comparing their magneto-optical data with the present work (e.g., in Fig. 4) would strengthen the completeness and impact of the discussion.

Overall, this is an outstanding piece of work with high originality and broad implications for the field. I strongly recommend this work for publication in Nature Communications.

Version 1:

Reviewer comments:

Reviewer #1

(Remarks to the Author)

The revised manuscript and the authors' replies address many of my concerns, but the central issue of the microscopic origin of the observed MOKE remains unresolved. The authors continue to attribute the observed MOKE to a scalar-spin-chirality (topological) mechanism and propose that the different field dependence of the Hall effect and MOKE arises from a change in frequency dependence across the metamagnetic transition — yet no experimental evidence for such frequency dependence is presented. I am seriously concerned that a spin-orbit-coupling (SOC)-driven mechanism, which is discussed for Mn₃Sn, not magnetization induced MOKE, may be contributing, either instead of or in addition to a topological MOKE. In the present material, the observed Kerr signal (and even Hall effect) could therefore be a superposition of three contributions: (i) topological MOKE from scalar spin chirality, (ii) Mn₃Sn-type MOKE arising from TRS breaking specific to the antiferromagnetic order (but not directly proportional to net magnetization), and (iii) the ordinary magnetization-induced MOKE. The authors must explicitly separate (or convincingly argue why they can be neglected) these contributions. Without such discussion, their main claim that the MOKE is dominated by topological spin chirality is not substantiated.

Reviewer #2

(Remarks to the Author)

The authors have performed calculations demonstrating that the contribution of the conventional mechanism to the overall MOKE signal is less than 1%. In addition, they have included new magnetic force microscopy (MFM) data in an effort to rule out any signal contribution from domain walls. It appears that the authors have addressed the issues I raised in a constructive and appropriate manner. Therefore, I have no further questions regarding this paper and recommend it for publication.

Reviewer #3

(Remarks to the Author)

I have carefully reviewed the revised manuscript and the authors' responses to all reviewers. In my opinion, their revisions satisfactorily address the previous concerns. Considering that this work presents the first robust experimental demonstration of topological magneto-optical effects, it fully merits publication in a high-impact journal such as Nature Communications. Therefore, I recommend the manuscript for publication in its current form.

Version 2:

Reviewer comments:

Reviewer #2

(Remarks to the Author)

Dear referees:

Thank you for your professional assessments and insightful suggestions. Attached below are the summary of changes and point-to-point responses to comments and questions.

-Jing Xia, on behalf of all authors

Summary of Changes

Changes in the manuscript are highlighted in **yellow**.

1. Following Nature Communications format, I have moved extended data figs 1-4 from the main text to the supplementary information and renamed them as supplementary figs 1-4, organized the main text in sections of “Introduction”, “Results”, and “Discussions”, and removed inserted figures but kept figure captions in the main text.
2. In the supplementary information, I have added sections “MOKE after Zero-field Cool (ZFC)”, “Magnetic Force Microscopy (MFM)”, “Energy-dispersive X-ray spectroscopy (EDX)”, and “Estimating the Contribution of Magnetization to the MOKE Signal”.
3. In the “Methods”, I have added “Low-temperature magnetic force microscopy (MFM)” and moved the bulk of “Sagnac MOKE measurements” to the supplementary information.
4. In the “Acknowledgements”, I have added “We thank J.-G. Zheng for Energy-dispersive X-ray spectroscopy (EDX) at UC Irvine.”.
5. On page 3, I added “Note that the real material $\text{Co}_{1/3}\text{TaS}_2$ may deviate from this ideal triple-Q state, and the observed zero-field magnetization could include some spin contribution.”
6. On page 5, I added “This also confirms that the zero-field MOKE signal does not originate from uncompensated spins at AFM domain walls. If that were the case, ZFC, which produces a higher density of domain walls, would yield a larger MOKE signal than field cooling (FC), contrary to our observations. The negligible magnetic contribution from AFM domain walls is also supported by MFM measurements after ZFC (Supplementary Fig. 8), which show no detectable fringing fields from domain walls.”
7. On page 5, I added “We estimate that the magnetization contribution to the MOKE signal is negligible, less than 1% (see Supplementary Information).”.
8. On page 6, I added “We found that higher Co content leads to lower red-light reflectivity, an effect which we expect to persist at 1550 nm in the Sagnac measurements. (Supplementary Fig. 9c, d).”.
9. On page 9, I added “Similarly, small ratios are found in skyrmion lattices in CrVI_6 51 ($\theta_K \sim 2 \times 10^{-4}$ rad, $M \sim 3 \mu_B/\text{Cr}$) and graphene/ Fe_3GeTe_2 /graphene heterostructure 52 (RMCD $\sim 1 \times 10^{-3}$ rad, $M \sim 2.5 \mu_B/\text{Fe}$) primarily due to their large ferromagnetic moments.”.
10. On page 11, I added “Data availability: Source data are provided with this paper. They have been deposited in a figshare repository with (link TBA).”.

Response to Referees

(Responses are in blue for clarity.)

Reviewer #1 (Remarks to the Author):

Farhang et al. report the observation of the so-called topological magneto-optical Kerr effect (MOKE) in a compensated antiferromagnet. Using a Sanac interferometer microscope, they discuss a magneto-optical response arising from spin topology and further argue a correlation between Co composition and MOKE. The experiments appear carefully conducted, and the discussion on composition is intriguing. Although the topological MOKE must be the important topic, I have several concerns regarding the interpretation of MOKE and its novelty, which should be addressed before the manuscript can be reconsidered:

We thank the reviewer for his/her professional assessments and suggestions, and for acknowledging the importance of the topic discussed in our paper.

(i) The authors ascribe the zero-field magnetization to an orbital moment induced by a fictitious field in Line 93. Could you clarify why this assignment is justified?

This is a general statement for the tetrahedral triple-Q state, as I quote below from Ref. 29 (Nature Communications, 14(1):8346 (2023)): "Notably, this ordering does not have any net spin magnetization. However, the emergent magnetic field couples to the orbital degrees of freedom of the conduction electrons, giving rise to a uniform orbital magnetization and a large topological Hall effect characterized by scalar spin chirality."

The real material $\text{Co}_{1/3}\text{TaS}_2$, however, could deviate from an ideal tetrahedral triple-Q state, leading to a small spin contribution to the zero-field magnetization, but this doesn't change the discussion or the conclusion of this paper. We have added the following to page 3:

"Note that the real material $\text{Co}_{1/3}\text{TaS}_2$ may deviate from this ideal triple-Q state, and the observed zero-field magnetization could include some spin contribution."

(ii) The authors emphasize the SOC-free mechanism for the MOKE, but I am not convinced their data provide evidence. They show that a simple spin model can explain the anomaly in the magnetic-field dependence of MOKE, but similar anomalies could also arise from SOC-induced MOKE. Whenever the antiferromagnetic structure is modified by an external field, one would expect anomalies in MOKE.

For SOC-driven MOKE, the MOKE signal is expected to scale with the product of the SOC strength and the magnetization. However, as shown in our response to point (vii), this SOC-induced contribution is estimated to be less than 1% of the observed MOKE change across the 4–6 T metamagnetic transition, and less than 0.5% during the 0–4 T field sweep. Thus, SOC-induced MOKE can account for at most ~1% of the observed "anomalies."

Moreover, the field dependence of MOKE differs from that of the topological Hall effect in their data. If both phenomena share the same microscopic origin, their field dependence should be consistent. This discrepancy is critical, and the authors must clarify why the two responses behave differently.

Indeed, there is an opposite change between MOKE and DC Hall effect across the metamagnetic transition (4-6 T). While both MOKE and the Hall effect should scale with the fictitious U(1) gauge field $b_f \propto t_3 \chi_{ijk} \hat{n}$, i.e. $\theta_K = A_{MOKE} b_f$, $H_{Hall} = B_{Hall} b_f$, the proportionality factors A_{MOKE} and B_{Hall} are definitely different due to the frequency dependence. This can be seen when calculating the magneto-optical conductivity using the Kubo formula (Nature Communications, 11(1):118. (2020)).

$$\sigma_{xy}(\omega) = \hbar e^2 \int \frac{d^3k}{(2\pi)^3} \sum_{n \neq n'} (f_{n\mathbf{k}} - f_{n'\mathbf{k}}) \times \frac{\text{Im} \left[\langle \psi_{n\mathbf{k}} | v_x | \psi_{n'\mathbf{k}} \rangle \langle \psi_{n'\mathbf{k}} | v_y | \psi_{n\mathbf{k}} \rangle \right]}{(\epsilon_{n\mathbf{k}} - \epsilon_{n'\mathbf{k}})^2 - (\hbar\omega + i\eta)^2},$$

Within a single magnetic phase, one can assume that both A_{MOKE} and B_{Hall} remain unchanged, and expect that MOKE and the Hall effect would scale with each other. However, across a metamagnetic transition into a different magnetic phase, A_{MOKE} and B_{Hall} necessarily change differently as they are at different frequencies ω . As a result, the changes of θ_K and Hall can be different across the metamagnetic transition at 4-6 T. This is why we wrote on page 6 the following:

“Lastly, we note the DC Hall effect σ_{xy} (Fig. 2e) shows an opposite change across the metamagnetic transition, suggesting a rather complex frequency dependence related to the corresponding change of spin texture across the metamagnetic transition.”

(iii) Related to this point, I strongly recommend the authors measure the wavelength dependence. There should be the particular wavelength where the MOKE shows the same field dependence as the Hall conductivity, which is the genuine topological MOKE.

As discussed above, in the long-wavelength limit, such as at microwave frequencies, the proportionality factors A_{MOKE} and B_{Hall} are expected to evolve similarly across the metamagnetic transition. Correspondingly, the MOKE response at microwave wavelength should exhibit the same relative change as the Hall conductivity between 4 and 6 T. Unfortunately, we currently lack the capability to perform wavelength-dependent measurements to confirm this directly.

(iv) The contrast in the optical reflectivity image is attributed to variations in Co composition. Is there direct evidence supporting this assignment? The authors should take an EDX image in the corresponding region.

Our working knowledge is that the reflectivities of Co_xTaS_2 crystals with different Co-compositions are different. It is difficult to perform energy dispersive X-ray spectroscopy (EDX) chemical mapping in the same region of Sagnac imaging, as (1) both are time-consuming and cover a very small area on the sample surface, and (2) the surface is largely topographically featureless, lacking “landmarks”. This is why we didn’t provide direct evidence in the original submission for the correlation between optical reflectivity and Co composition.

During the revision, we have found a workaround: Since no known optical transitions exist in $Co_{1/3}TaS_2$ between red light (620 - 750 nm) and 1550 nm, we may assume that the optical reflectivity at 1550 nm qualitatively follows that at red wavelengths. Based on this assumption, we use a faster and wide-field laboratory microscope to measure red-light reflectivity in the same region where EDX was conducted, and infer qualitatively the corresponding 1550 nm reflectivity. Locating the same region in the lab-microscope after EDX measurements is made possible because there is a triangular pattern of several large pits serving as “landmarks”. We found that higher Co content leads to lower red-light reflectivity, an effect which we expect to persist at 1550 nm in the Sagnac measurements. (Supplementary Fig. 9c, d).

In the supplementary information, we have added a new section “Energy-dispersive X-ray spectroscopy (EDX)” to elaborate on the details. And on page 6 of the main text, we have added “We found that higher Co content leads to lower red-light reflectivity, an effect which we expect to persist at 1550 nm in the Sagnac measurements. (Supplementary Fig. 9c, d).”

(v) The authors argue a clear correlation between Co composition and the magnitude of MOKE. The 5 % change in reflectivity results in dramatic change of MOKE signal as shown in Fig. S1. What could be the underlying mechanism? Does the magnetic structure or spin-orbit coupling strength change with Co substitution?

The anomalous Hall effect (AHE), arising from spin chirality, has only been observed when the Co composition falls below a certain threshold, as reported in Ref. 47 (Physical Review B 109, L060403 (2024)). In our

measurements, regions with low optical reflectivity, corresponding to higher Co composition, exhibit a strongly suppressed MOKE signal. Although I don't have evidence for a precise mechanism, I suspect that electron or hole doping from varying Co content modifies the electronic structure, which in turn alters both the magnetic ground state and optical reflectivity. As the reviewer correctly pointed out, the MOKE signal changes very abruptly as a function of reflectivity (or Co composition). Such abruptness is unlikely to arise from a change in the spin-orbit coupling strength, which should vary smoothly with Co composition. Instead, our observation of an abrupt change of MOKE is more consistent with a magnetic phase transition driven by varying Co composition.

(vi) Topological MOKE has already been reported in several other systems. If the novelty of the present work lies in demonstrating the SOC-free mechanism, its significance is limited. In addition, the authors should provide a balanced discussion of prior studies especially in the introduction and clearly highlight what is genuinely new in their results.

“Topological MOKE” of the same name has been reported in several ferromagnetic systems that host skyrmion lattices, such as Gd₂PdSi₂ (Ref. 50), CrV₁₆ (Ref. 51), and graphene/Fe₃GeTe₂/graphene heterostructure (Ref. 52), where Ref. 51 and 52 are pointed out by reviewer # 3. Although sharing the same name “Topological MOKE”, these are rather different from the “Topological MOKE” in the theory paper Nature Communications, 11(1):118. (2020), which discusses antiferromagnetic systems instead of ferromagnetic systems and motivated our work. Another major difference is that the “Topological MOKE” contributions in those skyrmion lattices in ferromagnets only appear in a narrow range of magnetic fields, appearing as “bump features”, while in the antiferromagnetic system (this work), the “Topological MOKE” is everywhere, including at zero magnetic field.

In the “Discussion”, we have included the following, where the change is marked by yellow highlights: “The skyrmion lattice phase in Gd₂PdSi₂⁵⁰ has a topological contribution to MOKE from swirling spins, though limited to a narrow field range and a large net moment of $4 \mu_B$, resulting in a small ratio $\sim 7 \times 10^{-5} \text{ rad}/\mu_B$. Similarly, small ratios are found in skyrmion lattices in CrV₁₆⁵¹ ($\theta_K \sim 2 \times 10^{-4} \text{ rad}$, $M \sim 3 \mu_B/\text{Cr}$) and graphene/Fe₃GeTe₂/graphene heterostructure⁵² ($RMCD \sim 1 \times 10^{-3} \text{ rad}$, $M \sim 2.5 \mu_B/\text{Fe}$) primarily due to their large ferromagnetic moments.”

(vii) Even if the MOKE is primarily driven by spin topology, the spin-orbit coupling inevitably plays some role in real materials. It is therefore misleading to stress a purely SOC-free mechanism.

I don't think it is misleading to stress a SOC-free mechanism, as the SOC-induced contribution, which is proportional to the product of SOC and magnetization, is estimated to be less than 1% below:

This can be seen by comparing the magnetization M_z (Fig. 2c) and MOKE signal (Fig. 2d, red) during a magnetic field sweep from 0 to 4 T. At zero-field, $M_z(0 \text{ T}) = 0.01 \mu_B$, $\theta_K(0 \text{ T}) = 195 \mu\text{rad}$; at 4 Tesla just below the metamagnetic transition, the magnetization increases to $M_z(4 \text{ T}) = 0.10 \mu_B$, and MOKE only changes slightly to $\theta_K(4 \text{ T}) = 196 \mu\text{rad}$. This indicated that the MOKE due to magnetization is no bigger than $(196 - 195) \mu\text{rad} / (0.10 - 0.01 \mu_B) = 11 \mu\text{rad}/\mu_B$. Thus, even at 4 Tesla when the magnetization is $0.10 \mu_B$, the MOKE contribution from magnetization is at most $11 \mu\text{rad}/\mu_B * 0.10 \mu_B = 1.1 \mu\text{rad}$, i.e. **only 0.5 %** of the total $195(6) \mu\text{rad}$ signal. A similar argument applies to the field sweep across the metamagnetic transition between 4 T and 6 T (Figs. 2c and 2d). In this region, both magnetization M_z and MOKE θ_K increase sharply, by $0.08 \mu_B$ and $60 \mu\text{rad}$, respectively, due to a sudden reconstruction of the spin configuration (i.e., spin chirality). If the MOKE change were driven primarily by the SOC-magnetization mechanism, it would amount to only $11 \mu\text{rad}/\mu_B * 0.08 \mu_B = 0.9 \mu\text{rad}$, i.e., **just 1%** of the observed $60 \mu\text{rad}$. Therefore, the dominant origin of MOKE is clearly the change in spin chirality, not the net magnetization M_z during the metamagnetic transition.

Accordingly, we have added in the supplementary information a section “Estimating the Contribution of Magnetization to the MOKE Signal”. And in the main text page 5, we have added a sentence “**We estimate that the magnetization contribution to the MOKE signal is negligible, less than 1% (see Supplementary Information).**”.

(viii) Figure S4 shows that Mn_3Sn exhibits the largest ratio of MOKE to magnetization. Then, what is the specific advantage of the topological MOKE inferred from the present data?

The MOKE in Mn_3Sn (non-collinear but coplanar AFM) arises from a very different mechanism requiring SOC, and hence also different implications. By contrast, our results demonstrate that large MOKE can arise without requiring strong SOC, possible in materials composed of lighter elements, thus widening the material choice for spintronics. Notably, the MOKE/magnetization ratio in $Co_{1/3}TaS_2$ reported here is already 1/3 of that in Mn_3Sn , underscoring the strong potential of this SOC-independent mechanism.

Reviewer #2 (Remarks to the Author):

In this study, the authors investigate the magneto-optical Kerr effect (MOKE) in the non-coplanar antiferromagnet $Co_{1/3}TaS_2$. They propose that the observed MOKE signal is topological in nature and arises from a mechanism independent of spin-orbit coupling (SOC) and net magnetization. By employing Sagnac interferometry-based MOKE microscopy, the authors simultaneously resolve chiral domain structures and superparamagnetic transition behavior. The work compellingly argues that, in contrast to the conventional SOC-dependent MOKE mechanisms that require either relativistic spin-orbit coupling or a finite magnetization, the signal here originates from collective topological properties of compensated spins within a real-space spin texture. The findings are intriguing and the data are in high quality. Although the experimental observations are of significant interest, the authors' conclusions require further substantiation. Several concerns regarding the interpretation and experimental design should be addressed prior to publication.

We thank the reviewer for his/her professional assessments and suggestions, and for acknowledging the “significant interest” of our observations.

1. The authors attribute the MOKE signal in $Co_{1/3}TaS_2$ to a mechanism that does not involve SOC or net magnetization. However, certain experimental details appear to indicate a correlation between the MOKE signals and magnetic field-induced changes in magnetization. For example, a comparison between Figures 1b and 1d reveals an increase in the MOKE signal concomitant with a rise in the magnetization M_z . Moreover, an upturn in M_z is observed as the magnetic field is swept from 4 T to 6 T (Figure 2c) due to a transition from a triple-Q to a single-Q state, which is mirrored by a corresponding increase in the MOKE signal (Figure 2d). This suggests that at least part of the MOKE signal may stem from field-induced magnetization. In addition to the data collected during ZFW after 0.3 T FC, could the authors provide MOKE measurements after ZFC?

Certainly! The ZFC data is now presented in Supplementary Fig. 7b in the newly added Supplementary Information section “MOKE after Zero-field Cool (ZFC)”, which shows 100 times smaller MOKE signal compared to both FC (Supplementary Fig. 7a blue) and the subsequent ZFW (Supplementary Fig. 7b red), at the same location. The 100 times smaller MOKE signal in ZFC is due to cancellations between oppositely oriented chirality domains. We have added the reference of Supplementary Fig. 7b to the following sentence on page 5 in the main text:

“We observed negligible spontaneous θ_K after zero-field cool (ZFC) (Supplementary Fig. 7b), indicating the formation of oppositely polarized domains smaller than the optical beam size in the absence of a training field.”

We agree with the reviewer that some portion of the MOKE signal may originate from field-induced magnetization, which is the “conventional mechanism” for MOKE, but we demonstrate below that this contribution is extremely small, less than 1% of the full MOKE signal.

This can be seen by comparing the magnetization M_z (Fig. 2c) and MOKE signal (Fig. 2d, red) during a magnetic field sweep from 0 to 4 T. At zero-field, $M_z(0\text{ T}) = 0.01\ \mu_B$, $\theta_K(0\text{ T}) = 195\ \mu rad$; at 4 Tesla just below the metamagnetic transition, the magnetization increases to $M_z(4\text{ T}) = 0.10\ \mu_B$, and MOKE only changes slightly to $\theta_K(4\text{ T}) = 196\ \mu rad$. This indicated that the MOKE due to magnetization is no bigger than $(196 - 195)\ \mu rad / (0.10 - 0.01\ \mu_B) = 11\ \mu rad / \mu_B$. Thus, even at 4 Tesla when the magnetization is $0.10\ \mu_B$, the MOKE contribution from magnetization is at most $11\ \mu rad / \mu_B * 0.10\ \mu_B = 1.1\ \mu rad$, i.e. **only 0.5 %** of the total $195(6)\ \mu rad$ signal.

A similar argument applies to the field sweep across the metamagnetic transition between 4 T and 6 T (Figs. 2c and 2d). In this region, both magnetization and MOKE increase sharply, by $0.08\ \mu_B$ and $60\ \mu rad$, respectively, due to a sudden reconstruction of the spin configuration (i.e., spin chirality). If the MOKE change were

driven primarily by magnetization, it would amount to only $11 \mu\text{rad}/\mu_B * 0.08 \mu_B = 0.9 \mu\text{rad}$, i.e., **just 1%** of the observed $60 \mu\text{rad}$. Therefore, the dominant origin of MOKE is clearly the change in spin chirality, not the net magnetization during the metamagnetic transition.

Now consider the simultaneous increase of magnetic susceptibility (Fig. 1b) and zero-field MOKE (Fig. 1d) noticed by the reviewer. Both effects reflect the buildup of spin chirality, but one does not cause the other; instead, they are parallel consequences of the same underlying evolution of spin texture.

We thank the reviewer for bringing this up, which we didn't explain clearly in the original submission. Accordingly, we have added in the revised supplementary information a section "Estimating the Contribution of Magnetization to the MOKE Signal". And in the main text page 5, we have added the following sentence:

"We estimate that the magnetization contribution to the MOKE signal is negligible, less than 1% (see Supplementary Information).".

2. Unlike SQUID, which probes bulk magnetization, Signac MOKE is a local measurement technique. Thus, the detected signal may originate not only from the intrinsic crystal properties but also from magnetization at domain walls. Owing to the high sensitivity of Sagnac interferometry microscopy, the authors clearly visualize localized structures in the antiferromagnetic material, as shown in Figures 3 and 4. These images reveal the presence of pronounced magnetic domains within Co1/3TaS2. At domain boundaries, even in the absence of an external magnetic field ($B=0$), uncompensated magnetization can arise due to spontaneous moments, which could contribute to a MOKE signal at $B = 0$. Therefore, could part of the zero-field MOKE signal be attributable to incomplete spin compensation at domain walls, while the field-dependent component arises from field-induced magnetization (M_z)? To rule out contributions from both uncompensated moments and field-induced magnetization, it is recommended that the authors perform additional local magnetic measurements, such as magnetic force microscopy (MFM), to corroborate their interpretations.

We thank the reviewer for raising the very important question regarding domain walls. And below, we use Sagnac data and newly added magnetic force microscopy (MFM) data to rule out this scenario. Accordingly, we have added two sections in the Supplementary Information, "MOKE after Zero-field Cool (ZFC)", and "Magnetic Force Microscopy (MFM)". And in the main text on page 5, we have added the following: "This also confirms that the zero-field MOKE signal does not originate from uncompensated spins at AFM domain walls. If that were the case, ZFC, which produces a higher density of domain walls, would yield a larger MOKE signal than field cooling (FC), contrary to our observations. The negligible magnetic contribution from AFM domain walls is also supported by MFM measurements after ZFC (Supplementary Fig. 8), which show no detectable fringing fields from domain walls.".

Sagnac data: in Supplementary Fig. 7b in the newly added Supplementary Information section "MOKE after Zero-field Cool (ZFC)", we compare Sagnac measurements at the same location between field-cooling (FC) and zero-field-cooling (ZFC). The 0.3 T FC (blue) and subsequent zero-field warm (ZFW, red) traces in Supplementary Fig. 7a exhibit a large θ_K of $180 \mu\text{rad}$. In contrast, the ZFC measurement (blue in Supplementary Fig. 7b) yields a much smaller θ_K of $-2 \mu\text{rad}$. If the zero-field MOKE signal is from incomplete spin compensation at domain walls, we would expect a much smaller MOKE signal after FC because it has nearly zero domain walls. Instead, it shows a 100 times larger signal compared to ZFC, ruling out this scenario.

MFM data: We have conducted extensive low-temperature MFM measurements under various experimental conditions. The newly added Supplementary Fig. 8 shows a representative scan taken at 5 K after zero-field cooling, where AFM domains are expected to be most abundant. However, no magnetic domains or domain walls were detected within the MFM sensitivity. This indicates that the fringing fields produced by uncompensated moments at AFM domain walls are too weak to be resolved by MFM. This is consistent with the picture that the MOKE signal is (dominantly) due to the chirality of AFM domains instead of uncompensated magnetic moments at AFM domain walls.

Reviewer #3 (Remarks to the Author):

This work focuses on realizing a magneto-optical Kerr effect (MOKE) in the spin-compensated noncoplanar antiferromagnet Co1/3TaS2 without relying on spin-orbit coupling, thereby addressing a central open question in the

fields of magnetism and light–matter interactions. By employing a highly sensitive zero-loop Sagnac interferometer microscope, the authors achieved direct imaging of scalar spin chirality domains and their reversal driven by noncoplanar spin configurations. They report a giant spontaneous Kerr signal of up to 250 μrad under an applied magnetic field, at a technologically relevant telecommunication wavelength, which is comparable in magnitude to SOC-driven MOKE in noncollinear antiferromagnets (e.g., Mn3Sn). Importantly, the study also captures the phase transition between 1Q and 3Q magnetic states, revealing that the Kerr signal vanishes in the 1Q phase and emerges robustly in the 3Q phase. Furthermore, the authors provide compelling evidence of chirality domain dynamics, disentangle extrinsic influences such as local strain and defects, and carefully correlate the Kerr response with chemical inhomogeneity. This clear correspondence between spin texture and magneto-optical response underscores the reliability of the experimental observations. The experimental design is meticulous, the methodology is state-of-the-art, and the results constitute the first robust experimental demonstration of the theoretically predicted topological magneto-optical effects.

The significance of this work lies in its demonstration that large, macroscopically measurable "topological" MOKE signals can arise without relying on either spin–orbit coupling or band exchange splitting, thereby fundamentally broadening the current understanding of magneto-optical phenomena. By establishing a direct connection between scalar spin chirality and the optical response—most notably the disappearance of the Kerr signal in the 1Q and paramagnetic phases and its robust reemergence in the 3Q phase—the study provides a well-defined benchmark for identifying and manipulating topological magneto-optical effects. These findings mark a milestone in exploring topological light–matter interactions and open promising avenues for spintronics and opto-spintronics based on chiral magnets, with potential implications for ultrafast, energy-efficient, and stray-field-immune device technologies.

We thank the reviewer for his/her professional assessments and suggestions, and for acknowledging that our “results constitute the first robust experimental demonstration of the theoretically predicted topological magneto-optical effects”, and that they “mark a milestone” in this field.

Before publication, I would like to raise two minor comments for the authors’ consideration:

1) In the estimation of the scalar spin chirality ($\chi_{\{ijk\}}$), only the bottom triangular plaquette is considered. However, from the crystal structure, there appear to be two layers of spin textures. How does the top triangular plaquette contribute to the scalar spin chirality, and can it qualitatively change the relationship between magnetization and the Kerr signals?

We agree with the reviewer that there is more than one plaquette in the magnetic unit cell. In a “complete” theory, one would ideally perform a vector sum of each plaquette’s chirality multiplied by the corresponding hopping integral, with the direction of the resulting vector being along the normal direction of that plaquette. However, such a calculation is beyond the scope of this experimental paper, largely due to my lack of skill to calculate the hopping integral. From a symmetry point of view, however, such a vector sum would, to the first order, be proportional to the contribution from the single plaquette considered in this paper. Therefore, one could expect the same qualitative (and likely quantitative to the first order) relationship between magnetization and the Kerr signal by considering only one plaquette.

2) In the Discussion section, line 254, there seems to be a typo: Extended Data Fig. 2 should be corrected to Extended Data Fig. 4. In Extended Data Fig. 4, the work on the skyrmion lattice Gd2PdSi2 (Ref. 50) is cited and compared with the present results. However, for topological MOKE emerging in skyrmion lattices, two more recent studies should also be cited and discussed: Nat. Phys. 20, 1145–1151 (2024) and ACS Nano 18, 20055–20064 (2024). As this is a rapidly developing and competitive field, including these relevant references and, if possible, comparing their magneto-optical data with the present work (e.g., in Fig. 4) would strengthen the completeness and impact of the discussion.

We thank the reviewer for pointing these out. I have fixed the typo, and have added in the “Discussion” section (page 9) the following sentence with these two recent references on topological MOKE in skyrmion lattices:

“Similarly, small ratios are found in skyrmion lattices in CrVI₆⁵¹ ($\theta_K \sim 2 \times 10^{-4}$ rad, $M \sim 3 \mu_B/\text{Cr}$) and graphene/Fe₃GeTe₂/graphene heterostructure⁵² ($RMCD \sim 1 \times 10^{-3}$ rad, $M \sim 2.5 \mu_B/\text{Fe}$) primarily due to their large ferromagnetic moments.”

Overall, this is an outstanding piece of work with high originality and broad implications for the field. I strongly recommend this work for publication in Nature Communications.

We thank the reviewer again for his/her professional assessments.

Dear referees:

Thank you for your professional assessments and insightful suggestions. Attached below are the summary of changes and point-to-point responses to comments and questions.

-Jing Xia, on behalf of all authors

Summary of Changes

Changes in the manuscript are highlighted in yellow.

I have added the following clarifications to the revised manuscript:

Line 93: A clear correlation between the degree of spin non-coplanarity and the amplitude of spontaneous Hall signal was confirmed by temperature dependence of polarized neutron scattering³⁰.

Line 130: The pronounced temperature dependence of MOKE in $\text{Co}_{1/3}\text{TaS}_2$ stands in sharp contrast to the temperature-invariant behavior we observed in the coplanar non-collinear antiferromagnet Mn_3NiN ⁴⁶, underscoring their distinct origins.

Response to Referees

(Responses are in blue for clarity.)

Reviewer #1 (Remarks to the Author):

The revised manuscript and the authors' replies address many of my concerns, but the central issue of the microscopic origin of the observed MOKE remains unresolved. The authors continue to attribute the observed MOKE to a scalar-spin-chirality (topological) mechanism and propose that the different field dependence of the Hall effect and MOKE arises from a change in frequency dependence across the metamagnetic transition — yet no experimental evidence for such frequency dependence is presented. I am seriously concerned that a spin-orbit-coupling (SOC)–driven mechanism, which is discussed for Mn₃Sn, not magnetization induced MOKE, may be contributing, either instead of or in addition to a topological MOKE. In the present material, the observed Kerr signal (and even Hall effect) could therefore be a superposition of three contributions: (i) topological MOKE from scalar spin chirality, (ii) Mn₃Sn-type MOKE arising from TRS breaking specific to the antiferromagnetic order (but not directly proportional to net magnetization), and (iii) the ordinary magnetization-induced MOKE. The authors must explicitly separate (or convincingly argue why they can be neglected) these contributions. Without such discussion, their main claim that the MOKE is dominated by topological spin chirality is not substantiated.

We thank the reviewer for raising the question regarding the separation of the possible contributions (i), (ii), and (iii). As clarified in our previous revision, contribution (i) is at least two orders of magnitude larger than (iii). To further address mechanism (ii) “Mn₃Sn-type MOKE arising from TRS breaking specific to the antiferromagnetic order (but not directly proportional to net magnetization)”, we now cite two key experimental results, including one from our group, showing that mechanism (i) dominates while (ii) doesn't dominate in Co_{1/3}TaS₂.

(1) As the reviewer noted, mechanism (ii) gives rise to MOKE and AHE in coplanar noncollinear antiferromagnets such as Mn₃Sn, where the Hall conductivity σ_{xy} is given by the Berry curvature integral as $\sigma_{xy} = \frac{-e^2}{2\pi^3\hbar} \int_{BZ} dk \Omega(k)$. Here $\Omega(k) = \sum_n f[\epsilon_n(k) - \mu] \Omega_n(k)$ is the total Berry curvature over the occupied bands in the Brillouin zone. It is expected that once the AFM phase is fully developed, σ_{xy} should be largely temperature independent since the Berry curvature is a band structure property. Accordingly, the MOKE signal $\theta_K(\omega) =$

$\text{Re} \left[\frac{-\sigma_{xy}(\omega)}{\sigma_{xx}(\omega) \sqrt{1 + i \left(\frac{4\pi}{\omega} \right) \sigma_{xx}(\omega)}} \right]$ is also expected to be temperature invariant if mechanism (ii) dominates.

We have recently reported such behavior in our Sagnac-interferometer MOKE measurements on the coplanar noncollinear antiferromagnet Mn₃NiN (*Farhang et al., arXiv:2510.19709*), where the spontaneous Kerr signal remains nearly constant with temperature, consistent with a Berry-curvature-driven origin, as shown in the adopted figure above. In stark contrast, Co_{1/3}TaS₂ exhibits a strong temperature dependence in MOKE (Fig. 1d of the main text), clearly indicating that mechanism (ii) is not dominant in this system.

(2) A quantitative comparison between mechanism (i) and (ii) requires the experimental measurement of spin non-coplanarity, which directly corresponds to contribution (i) but is irrelevant to (ii). Such measurements in Co_{1/3}TaS₂ were reported in *Takagi, H. et al., Nat. Phys. 19, 961–968 (2023)*, where polarized neutron scattering

confirmed a clear correlation between the degree of spin non-coplanarity and the amplitude of the spontaneous Hall signal, quote “Supplementary Note IX discusses the temperature dependence of polarized neutron scattering data, which confirms the clear correlation between the degree of spin non-coplanarity and the amplitude of spontaneous Hall signal and Supplementary Note X provides a detailed discussion on the mechanism of Hall effect in antiferromagnets”. Their result establishes that mechanism (i) dominates the AHE in $\text{Co}_{1/3}\text{TaS}_2$. Since MOKE is the optical-frequency analog of the AHE, it follows that mechanism (i) is also the primary contributor to MOKE in $\text{Co}_{1/3}\text{TaS}_2$.

Accordingly, we have added the following clarifications to the revised manuscript:

Line 93: A clear correlation between the degree of spin non-coplanarity and the amplitude of spontaneous Hall signal was confirmed by temperature dependence of polarized neutron scattering³⁰.

Line 130: The pronounced temperature dependence of MOKE in $\text{Co}_{1/3}\text{TaS}_2$ stands in sharp contrast to the temperature-invariant behavior we observed in the coplanar non-collinear antiferromagnet Mn_3NiN ⁴⁶, underscoring their distinct origins.

Reviewer #2 (Remarks to the Author):

The authors have performed calculations demonstrating that the contribution of the conventional mechanism to the overall MOKE signal is less than 1%. In addition, they have included new magnetic force microscopy (MFM) data in an effort to rule out any signal contribution from domain walls. It appears that the authors have addressed the issues I raised in a constructive and appropriate manner. Therefore, I have no further questions regarding this paper and recommend it for publication.

We thank the reviewer again for his/her professional assessments.

Reviewer #3 (Remarks to the Author):

I have carefully reviewed the revised manuscript and the authors' responses to all reviewers. In my opinion, their revisions satisfactorily address the previous concerns. Considering that this work presents the first robust experimental demonstration of topological magneto-optical effects, it fully merits publication in a high-impact journal such as Nature Communications. Therefore, I recommend the manuscript for publication in its current form.

We thank the reviewer again for his/her professional assessments.

Dear referees:

Thank you for your professional assessments and insightful suggestions. Attached below are the summary of changes and point-to-point responses to comments and questions.

-Jing Xia, on behalf of all authors

Summary of Changes

Changes in the manuscript are highlighted in **yellow**.

There is no further change to the manuscript in this revision.

Response to Referees

(Responses are in blue for clarity.)

REVIEWERS' COMMENTS

R#2:

Reviewer 1 disputes the authors' attribution of the observed MOKE signal solely to a scalar spin chirality (topological) mechanism. He proposed that the MOKE observed in this study may also comprise two additional components: 1) a contribution arising from the time-reversal symmetry breaking specific to the antiferromagnetic order, and 2) the conventional magnetization-induced MOKE.

I had previously raised a similar concern regarding the potential influence of component 2). In their response, the authors provided detailed calculations, concluding that "we estimate the magnetization contribution to the MOKE signal is negligible, less than 1%." This justification satisfactorily addresses and effectively rules out the second factor.

However, Reviewer 1 points out that the MOKE generated via the spin-orbit coupling (SOC) driven mechanism is not directly proportional to the net magnetization, a phenomenon previously documented in Mn₃Sn [Nature Photonics 12, 73–78 (2018)]. As a plausible underlying reason for the observed signal, the authors need to provide a thorough discussion of this specific issue in their revised manuscript.

We thank the reviewer again for his/her professional assessment and for raising this crucial point, which echoes a question from Reviewer 1 in the last revision: whether the observed MOKE signal arises from a Spin-Orbit Coupling (SOC) driven mechanism that is non-proportional to net magnetization, similar to the phenomenon documented in Mn₃Sn [Nature Photonics 12, 73–78 (2018)].

As discussed in our response to Reviewer 1 in the last revision, the physical origin of the MOKE in the non-coplanar antiferromagnet Co_{1/3}TaS₂ (this work) is fundamentally different from that of the coplanar non-collinear antiferromagnet such as Mn₃Sn. In Co_{1/3}TaS₂, the MOKE arises from the Berry phase associated with the real-space scalar spin chirality of the non-coplanar spin arrangement (a topological effect not explicitly reliant on SOC). In contrast, the signal in non-collinear antiferromagnet Mn₃Sn results from the momentum-space Berry curvature of the magnetic band structure driven by SOC. This distinction is supported with two key pieces of evidence that we have previously explained in the last revision:

(1) **Temperature Dependence vs. Invariance:** In systems like Mn₃Sn, the intrinsic anomalous Hall effect (AHE) and MOKE are band structure properties. Therefore, once the AFM order is established, these signals should be largely temperature-independent. We recently verified this behavior by observing a "temperature-invariant" MOKE in the non-collinear antiferromagnet Mn₃NiN (Farhang et al., arXiv:2510.19709). In stark contrast, Co_{1/3}TaS₂ exhibits a strong temperature dependence in its MOKE signal. This is because the signal tracks the scalar spin chirality, which itself varies significantly with temperature in this system. This distinct behavior strongly suggests a chirality-driven origin rather than a purely SOC-driven band structure effect.

In the last revision, we have added the following clarifications in Line 93: A clear correlation between the degree of spin non-coplanarity and the amplitude of spontaneous Hall signal was confirmed by temperature dependence of polarized neutron scattering³⁰.

(2) **Correlation with Spin Chirality:** A quantitative link between spin non-coplanarity and the spontaneous Hall effect in Co_{1/3}TaS₂ has been established by Takagi et al. [Nat. Phys. 19, 961–968 (2023)]. Their polarized neutron scattering results confirmed that the amplitude of the spontaneous Hall signal is directly correlated with the degree of spin non-coplanarity, identifying scalar spin chirality as the dominant mechanism for the AHE. Since MOKE is the optical-frequency analog of the AHE, it follows that scalar spin chirality is also the primary contributor to the MOKE observed in our work.

In the last revision, we have added the following clarifications in Line 130: The pronounced temperature dependence of MOKE in $\text{Co}_{1/3}\text{TaS}_2$ stands in sharp contrast to the temperature-invariant behavior we observed in the coplanar non-collinear antiferromagnet Mn_3NiN ⁴⁶, underscoring their distinct origins.